# Variability in engagement and progress in efficacious integrated collaborative care for primary care patients with obesity and depression: Within-treatment analysis in the RAINBOW trial

Nan Lv[1], Lan Xiao[2], Marzieh Majd[3], Philip W. Lavori[4], Joshua M. Smyth[3], Lisa G. Rosas[5], Elizabeth M. Venditti[6], Mark B. Snowden[7], Megan A. Lewis[8], Elizabeth Ward[9], Lenard Lesser[10], Leanne M. Williams[11], Kristen M. J. Azar[12], Jun Ma[1,13]*

1 Institute of Health Research and Policy, University of Illinois at Chicago, Chicago, Illinois, United States of America, 2 Department of Medicine, Stanford University, Palo Alto, California, United States of America, 3 Department of Biobehavioral Health, Pennsylvania State University, University Park, Pennsylvania, United States of America, 4 Department of Biomedical Data Science, Stanford University, Stanford, California, United States of America, 5 Department of Health Research and Policy and Medicine, Stanford University, Palo Alto, California, United States of America, 6 Department of Psychiatry, University of Pittsburgh School of Medicine, Pittsburgh, Pennsylvania, United States of America, 7 Department of Psychiatry and Behavioral Sciences, University of Washington, Seattle, Washington, United States of America, 8 Center for Communications Science, RTI International, Seattle, Washington, United States of America, 9 Pacific Coast Psychiatric Associates, San Francisco, California, United States of America, 10 One Medical, San Francisco, California, United States of America, 11 Department of Psychiatry and Behavioral Sciences, Stanford University, Stanford, California, United States of America, 12 Sutter Health Research Enterprise, Center for Health Systems Research, Walnut Creek, California, United States of America, 13 Department of Medicine, University of Illinois at Chicago, Chicago, Illinois, United States of America

* maj2015@uic.edu

**Data Availability Statement:** To comply with the study informed consent form, we would share de-identified data and associated data dictionary only

## Abstract

### Introduction

The RAINBOW randomized clinical trial validated the efficacy of an integrated collaborative care intervention for obesity and depression in primary care, although the effect was modest. To inform intervention optimization, this study investigated within-treatment variability in participant engagement and progress.

### Methods

Data were collected in 2014–2017 and analyzed post hoc in 2018. Cluster analysis evaluated patterns of change in weekly self-monitored weight from week 6 up to week 52 and depression scores on the Patient Health Questionnaire-9 (PHQ-9) from up to 15 individual sessions during the 12-month intervention. Chi-square tests and ANOVA compared weight loss and depression outcomes objectively measured by blinded assessors to validate differences among categories of treatment engagement and progress defined based on cluster analysis results.

under a formal data sharing and use agreement that provides for a commitment to the following: (1) using the data only for research purposes and not to identify any individual participant, (2) securing the data using appropriate computer technology, which needs to be specified, (3) destroying or returning the data after analyses are completed, (4) accepting reporting responsibilities, (5) abiding by restrictions on redistribution of the data for commercial purposes or to third parties, and (6) proper acknowledgement of the data resource. Data sharing request shall be submitted to the Institutional Review Board for the University of Illinois at Chicago whose contact information is below. Telephone: 1-(312) 996-1711 Email: uicirb@uic.edu".

**Funding:** "Research reported in this publication was supported by the National Heart, Lung, And Blood Institute of the National Institutes of Health (URL: https://www.nhlbi.nih.gov/) under Award Number R01HL119453 (Recipient: JM). The funder had no role in study design, data collection and analysis, decision to publish, or preparation of the manuscript. The content is solely the responsibility of the authors and does not necessarily represent the official views of the National Institutes of Health. There were no other funding sources."

**Competing interests:** The authors declare that no competing interests existed for the research as reported. Dr. Lenard Lesser's affiliation with 1Life Healthcare/One Medical constituted no competing interests. Dr. Lesser had originally supported on the NIH grant for his role as a study physician on this study while he was an employee at the Palo Alto Medical Foundation Research Institute (PAMFRI) where the study was conducted. Starting from 07/2016 till 10/2017, Dr. Lesser transitioned from PAMFRI to 1Life Healthcare, Inc/One Medical. At that time, his continuous involvement in the study was compensated through a research contract as an independent consultant with PAMFRI specifically for the NIH grant supporting the study. 1Life Healthcare/One Medical provided no support of any form for the study; and had no role in the study design, data collection/analysis, decision to publish, or preparation of the manuscript. This does not alter our adherence to PLOS ONE policies on sharing data and materials."

## Results

Among 204 intervention participants (50.9 [SD, 12.2] years, 71% female, 72% non-Hispanic White, BMI 36.7 [6.9], PHQ-9 14.1 [3.2]), 31% (n = 63) had poor engagement, on average completing self-monitored weight in <3 of 46 weeks and <5 of 15 sessions. Among them, 50 (79%) discontinued the intervention by session 6 (week 8). Engaged participants (n = 141; 69%) self-monitored weight for 11–22 weeks, attended almost all 15 sessions, but showed variable treatment progress based on patterns of change in self-monitored weight and PHQ-9 scores over 12 months. Three patterns of weight change (%) represented minimal weight loss (n = 50, linear $\beta_1$ = -0.06, quadratic $\beta_2$ = 0.001), moderate weight loss (n = 61, $\beta_1$ = -0.28, $\beta_2$ = 0.002), and substantial weight loss (n = 12, $\beta_1$ = -0.53, $\beta_2$ = 0.005). Three patterns of change in PHQ-9 scores represented moderate depression without treatment progress (n = 40, intercept $\beta_0$ = 11.05, $\beta_1$ = -0.11, $\beta_2$ = 0.002), moderate depression with treatment progress (n = 20, $\beta_0$ = 12.90, $\beta_1$ = -0.42, $\beta_2$ = 0.006), and milder depression with treatment progress (n = 81, $\beta_0$ = 7.41, $\beta_1$ = -0.23, $\beta_2$ = 0.003). The patterns diverged within 6–8 weeks and persisted throughout the intervention. Objectively measured weight loss and depression outcomes were significantly worse among participants with poor engagement or poor progress on either weight or PHQ-9 than those showing progress on both.

## Conclusions

Participants demonstrating poor engagement or poor progress could be identified early during the intervention and were more likely to fail treatment at the end of the intervention. This insight could inform individualized and timely optimization to enhance treatment efficacy.

## Trial registration

ClinicalTrials.gov# NCT02246413.

## Introduction

Obesity and depression are highly prevalent in the United States with associated high personal and societal cost. [1, 2] Currently among US adults, nearly 40% are obese [3] and 19% experience major depression over the course of their lifetime. [4] Subthreshold depression is also common, with increased burden of morbidity and disability. [5, 6] Mounting epidemiologic evidence shows a temporally reciprocal, positive relationship between obesity and depression; namely, people with obesity are more likely to develop new-onset depression or have worsening depressive symptoms, and vice versa. [7–11]

The high co-occurrence of these 2 conditions reveals a critical need for developing effective multimorbidity treatment. Randomized clinical trials (RCTs) of integrated behavior therapy for patients with obesity and depression are limited and have shown mixed results. [12–14] Recently, the Research Aimed at Improving Both Mood and Weight (RAINBOW) trial reported that an integrated collaborative care intervention, as compared with usual care, led to significantly improved weight loss and depressive symptoms through 12 months among primary care patients of both sexes who had obesity and depression. [14] Similar to prior trials showing effectiveness of behavior therapy in either of these conditions alone [15–18] or in related multiple chronic conditions—such as depression and diabetes or

coronary heart disease [19]—the magnitude of treatment effects on both weight loss and depression outcomes in the trial were modest.

The modest effects may be caused by the variability in treatment engagement and progress, which is typically high in clinical settings. Examination of this variability can inform optimization—such as when and how to adapt intervention delivery or content for enhanced efficacy—of behavioral interventions. However, research on this topic is lacking, especially in multimorbidity management.

This study reports on post hoc analyses aimed to investigate variability in treatment engagement and progress during the integrated collaborative care intervention among RAINBOW patients with comorbid obesity and depression.

## Materials and methods

The Institutional Review Board for Sutter Health, Northern California, approved the study. All participants provided written informed consent. The trial protocol was published previously. [20] The co-primary efficacy outcomes were changes in body mass index (BMI) and Depression Symptom Checklist 20-item (SCL-20) [21, 22] scores objectively obtained by blinded outcome assessors. A total of 409 participants who had both BMI $\geq$30 ($\geq$27 if Asian) and Patient Health Questionnaire 9-item (PHQ-9) scores $\geq$10, and no exclusions per protocol, were enrolled in the trial. Participants were randomly assigned to the 12-month I-CARE (*I*ntegrated *Coa*ching fo*r* B*e*tter Mood and Weight) intervention group (n = 204) or the usual care control group (n = 205). This study analyzed participant data only within the intervention group.

### Intervention

The I-CARE intervention integrated a self-directed Group Lifestyle Balance™ (GLB) program for weight loss [23–25] and the Program to Encourage Active, Rewarding Lives for Seniors (PEARLS) program [26, 27] for collaborative stepped depression care. The GLB program [25] was adapted from the Diabetes Prevention Program [28] and provided videos for patient self-study. The PEARLS program used Problem-Solving Therapy (PST) combined with behavioral activation strategies as the first-line approach and, if indicated, therapy was intensified through stepwise increases in doses and number of antidepressant medications. The intervention had a 6-month intensive treatment phase comprising 9 one-on-one in-person visits of 60 minutes each, 11 home-viewed GLB videos of 20 to 30 minutes each, and digital self-monitoring activities; and a 6-month maintenance phase comprising 6 phone calls of 15 to 30 minutes each and continued self-monitoring. Participants met with a health coach weekly for the first 4 sessions, every 2 weeks for the next 2 sessions, and every month for the last 3 sessions; the maintenance phase included only monthly phone calls. Scheduling deviations were permissible to accommodate participant availability and preferences.

Participants received the PEARLS program for depression starting with the first in-person visit and were instructed to initiate the GLB video program after it was formally introduced during the fifth intervention session. The intervention outline is provided in S1 Appendix. A trained bachelor's-level health coach delivered the intervention, and a supervising master's-level registered dietitian oversaw fidelity assurance. They both met every 1 to 2 weeks with a psychiatrist and a primary care physician to review patient progress and discuss new and difficult cases. Additional detail on the intervention format, structure, and content and fidelity assurance procedures is provided in the published protocol. [20]

## Measures

Process data were collected to evaluate participants' progress over the year-long intervention during 2014–2017. After the GLB program was formally introduced in Week 6, participants were asked to manually enter their weight and minutes of leisure-time physical activity at least weekly using MyFitnessPal website or app. Also, participants were asked to wear a study-provided Fitbit pedometer that interfaced with a personal computer or the Fitbit app on a mobile device to automatically upload daily steps into the participant's Fitbit account. The health coach was able to review the person's self-tracked data, monitor their progress, and use it to facilitate individualized coaching during intervention sessions. In addition, the health coach administered the PHQ-9 at the beginning of each in-person or phone session. [29] Each participant could have up to 46 weeks with self-monitored weight, minutes of physical activity data, or steps as expected (from week 6 to week 52) and a maximum of 15 sessions (or 15 PHQ-9 scores). Indices of behavioral adherence to the intervention included the number of intervention sessions attended and the number of weeks with self-monitored weight, self-reported physical activity minutes, and FitBit steps separately.

Weight loss and depression outcomes used to validate the treatment engagement and progress categories in this study included weight loss and depression related primary and secondary outcomes objectively measured by blinded outcome assessors at baseline, 6, and 12 months in the RAINBOW trial. As primary outcomes, BMI was calculated as weight (kg) divided by height squared (m2); and depression severity was measured with the SCL-20 scores, ranging 0 (best) to 4 (worst). [22] Secondary outcome measures included ≥5% decrease in weight from baseline, [30] depression treatment response (i.e., ≥50% decrease in SCL-20 scores from baseline), [19, 26, 27] and complete depression remission (i.e., SCL-20 scores<0.5). [26, 27] Of 204 intervention participants, 196 and 183 had objectively-measured weight data at 6 and 12 months, respectively; and 175 and 169 had SCL-20 data at 6 and 12 months, respectively.

## Statistical analysis

**Cluster analysis on patterns of percent weight change and PHQ-9 score change.** - Patterns of change in 2 variables—percent weight change and PHQ-9 scores—were assessed separately using a method similar to the one by Babbin et al. [31] Both variables had direct relevance to treatment progress monitoring. The 1-year intervention period was examined in 4 quarters, and only participants who had any data in a quarter for at least 3 or all 4 quarters were included in the cluster analyses (n = 123/60% for self-monitored weight and n = 141/69% for PHQ-9, respectively). This approach was applied to enhance the reliability of change patterns during the yearlong intervention and reduce the influence of participants with missing data in 2 or more quarters. For either percent weight change or PHQ-9 scores, participants with no data in at least 3 quarters were classified as "cluster 0." Cluster analyses for both percent weight change and PHQ9 scores followed the same 3 steps. First, the k-means method in the SAS FASTCLUS procedure without pre-specification of the number of clusters was used to group participants who had at least 1 measurement in each of the 4 quarters into clusters of individuals with similar patterns of change over time based on their 4 quarterly means. This step produced different numbers of clusters (range 2–6). Second, the optimal number of clusters was determined using a combination of criteria, including Pseudo F statistic (a relatively large value), R-squared value (a peak that flattens with additional clusters), Cubic Clustering Criterion (≥2), and cluster size (≥10 participants). [32] The optimal number of clusters was 3 for both percent weight change and PHQ-9 scores. Third, participants with percent weight change data in any 3 of the 4 quarters were assigned to their closest cluster defined by the smallest of the Euclidean distances between a participant's 3 available quarterly means and

each cluster's means in the corresponding quarters. Using the same method, participants with PHQ-9 scores in any 3 of the 4 quarters were assigned to their closest cluster.

**Internal consistency and sensitivity analysis.** To compare individual trajectories within the resulting clusters, the polynomial regression was used to fit the trajectory of each participant's available data on percent weight change and PHQ-9 scores over the course of the intervention. Then, analysis of variance (ANOVA) was used to compare intercept (for PHQ-9 only), linear, and quadratic coefficients of the individual trajectories among the 3 clusters for percent weight change and PHQ-9 separately. We also tested whether the polynomial model for each cluster fit the data well using the significance of polynomial terms, adjusted $R^2$, and the Bayesian information criterion (BIC). For both percent weight change and PHQ-9 score change, the polynomial regression models with a quadratic term fit the data better than the ones without given the significance of the quadratic terms, higher adjusted $R^2$, and lower BIC. Hence, the final models included both linear and quadratic terms.

Additionally, we tested whether the clusters derived separately for percent weight change and PHQ-9 were concordant with the clusters derived jointly for both variables because the integrated intervention addressed both obesity and depression. To do this, a sensitivity analysis was conducted using participants who had data on both variables in all 4 quarters (n = 88). The k-means method was applied in the separate and joint cluster analyses of the 88 participants.

**Categorization and validation of treatment engagement and progress.** Based on a cross tabulation of clusters 0 to 3 of percent weight change and PHQ-9 scores separately, all 204 intervention participants were grouped into 3 categories of treatment engagement and progress: poor engagement, poor progress, and progress. The poor engagement category included participants who had poor session attendance (i.e., cluster 0 for PHQ-9). The poor progress category included participants who had minimal improvement in self-monitored weight or PHQ-9 (i.e., cluster 1 for either) or had poor self-monitoring of weight despite attending sessions (i.e., cluster 0 for weight and cluster 1, 2, or 3 for PHQ-9). The progress category included participants who had improvements in both self-monitored weight and PHQ-9 (i.e., cluster 2 or 3 for both). For validation, intervention adherence indices—such as the number of sessions attended and the number of weeks with self-monitoring data as well as objectively measured BMI and SCL-20 at 6 and 12 months—were compared among the treatment engagement and progress categories. ANOVA was used for continuous variables and the chi-square test was used for categorical variables.

All analyses were conducted in 2018 using SAS version 9.4 (SAS Institute Inc., Cary, North Carolina), except for sensitivity cluster analysis, which was conducted in kml and kml3d R packages. [33] Statistical significance was defined as $P<0.05$ (2-sided).

## Results

### Baseline characteristics

Baseline participant characteristics were previously published. [14] The 204 intervention participants were primarily middle aged (mean 50.9 [SD 12.2] years), female (71%), non-Hispanic White (72%), and at least college educated (70%) (Table 1). They had moderately severe obesity (BMI, mean 36.7 [SD 6.9]) and depression (PHQ-9, 14.1 [3.2]; SCL-20, 1.5 [0.5]).

### Clusters of percent weight change and PHQ-9 scores separately

Participants with self-monitored weight data in at least 3 quarters of the 12-month intervention period (n = 123) had similar baseline characteristics as the entire intervention group (Table 1). Among the 123 participants, the 3 clusters of percent weight change trajectories

**Table 1. Baseline characteristics of RAINBOW intervention participants (n = 204).**

| Characteristic | All I-CARE participants (n = 204) | Participants included in cluster analysis of percent weight change (n = 123) | Participants included in cluster analysis of PHQ-9 scores (n = 141) |
|---|---|---|---|
| Age, year | 50.9 (12.2) | 52 (11.6) | 51.2 (11.9) |
| Female, No. (%) | 144 (71) | 84 (68) | 100 (71) |
| Race/Ethnicity, No. (%) | | | |
| Non-Hispanic White | 147 (72) | 94 (76) | 105 (74) |
| Minority | 57 (28) | 29 (24) | 36 (26) |
| Education, No. (%) | | | |
| High school to some college | 61 (30) | 31 (25) | 39 (28) |
| College graduate | 78 (38) | 55 (45) | 61 (43) |
| Post college | 65 (32) | 37 (30) | 41 (29) |
| Income, No. (%), n = 176 | | | |
| <$100,000 | 66 (38) | 37 (35) | 46 (38) |
| $100,000- <$150,000 | 34 (19) | 21 (20) | 25 (21) |
| ≥$150,000 | 76 (43) | 47 (45) | 50 (41) |
| Marital status, No. (%), n = 203 | | | |
| Married/living with a partner | 123 (61) | 80 (66) | 84 (60) |
| Single/separated/divorced/widowed | 80 (39) | 42 (34) | 56 (40) |
| Household size, No. (%), n = 203 | | | |
| < 2 | 40 (20) | 20 (16) | 27 (19) |
| = 2 | 74 (36) | 48 (39) | 53 (38) |
| 3+ | 89 (44) | 55 (45) | 61 (43) |
| BMI, kg/m$^2$ | 36.7 (6.9) | 36.7 (7.0) | 36.9 (6.9) |
| PHQ-9 | 14.1 (3.2) | 13.7 (3.2) | 13.9 (3.2) |
| SCL-20 | 1.5 (0.5) | 1.4 (0.5) | 1.4 (0.5) |

Abbreviations: BMI, body mass index; PHQ-9, Patient Health Questionnaire-9; SCL20, Symptom Checklist-20.

Values are mean (SD) unless otherwise noted.

were as follows: (1) minimal weight loss (n = 50; $\beta_1$ = -0.06, $\beta_2$ = 0.001), (2) moderate weight loss (n = 61; $\beta_1$ = -0.28, $\beta_2$ = 0.002), and (3) most weight loss (n = 12; $\beta_1$ = -0.53, $\beta_2$ = 0.005) (Fig 1 and S2 Appendix **for plots of individual trajectories within each cluster**). Both the linear ($\beta_1$) and quadratic terms ($\beta_2$) were statistically different from zero for these clusters (all P<0.001). Pairwise comparisons of $\beta_1$ and $\beta_2$ coefficients of the individual trajectories across the 3 clusters showed that $\beta_1$s within cluster 3 were significantly lower than those within cluster 2, which were significantly lower than those within cluster 1; and $\beta_2$s within cluster 3 was not significantly different from those within cluster 2 but both were significantly higher than those within cluster 1 (S3 Appendix). Fig 1 shows that separation of the clusters began within 6 to 8 weeks and grew over time; mean weight loss reached 5% within 12 weeks in cluster 3, but not until beyond 20 weeks in cluster 2, and never in cluster 1.

Participants with PHQ-9 scores in at least 3 quarters (n = 141) had similar baseline characteristics as the entire intervention group (Table 1). Among these participants, the 3 clusters of PHQ-9 trajectories were as follows: (1) moderate depression without treatment progress (n = 40; $\beta_0$ = 11.05, $\beta_1$ = -0.11, $\beta_2$ = 0.002), (2) moderate depression with treatment progress (n = 20; $\beta_0$ = 12.90, $\beta_1$ = -0.42, $\beta_2$ = 0.006), and 3) milder depression with treatment progress

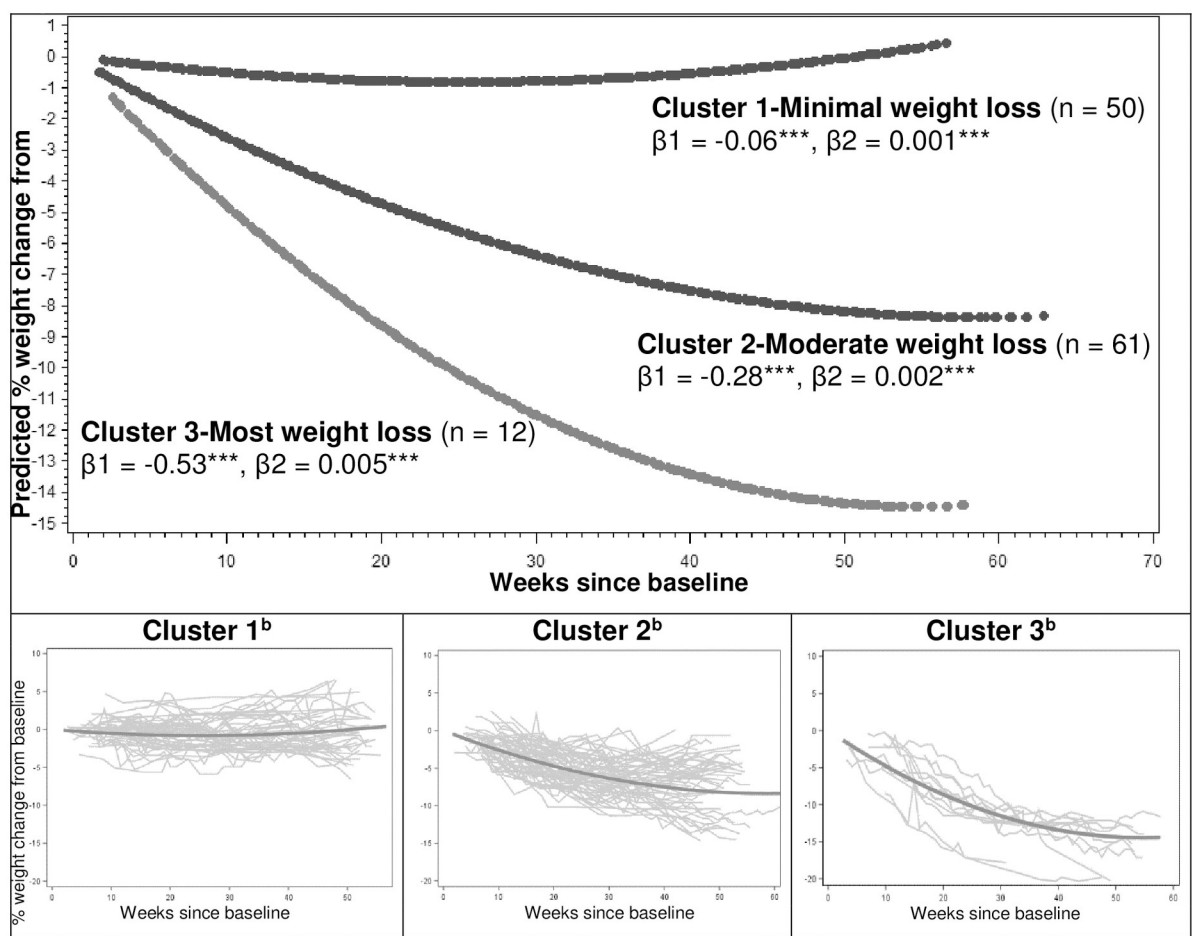

β1, linear coefficient; β2, quadratic coefficient. ***P<.001
[a]123 participants, or 60% of the intervention group (n=204), had self-monitored weight data in at least 3 quarters of the 12-month intervention period.
[b]Light gray lines show individual participant trajectories within each cluster.

**Fig 1. Percent weight change trajectories among intervention participants with self-monitored weight data in at least 3 quarters of the 12-month intervention period[a,b].** β1, linear coefficient; β2, quadratic coefficient. ***P < .001. [a]123 participants, or 60% of the intervention group (n = 204), had self-monitored weight data in at least 3 quarters of the 12-month intervention period. [b]Light gray lines show individual participant trajectories within each cluster.

(n = 81; β0 = 7.41, β1 = -0.23, β2 = 0.003) (Fig 2 and S4 Appendix **for plots of individual trajectories within each cluster**). The intercept (β0), linear term (β1), and quadratic terms (β2) were statistically significant from zero for these clusters (all P<0.001, except for cluster 1 β1, P<0.01 and β2, P<0.05). Additionally, pairwise comparisons of β0, β1, and β2 coefficients of the individual trajectories across the 3 clusters showed that β0s within cluster 2 were significantly higher than those within cluster 1, which were significantly higher than those within cluster 3; β1s within cluster 1 were significantly higher than those within cluster 3, which were significantly higher than those within cluster 2; and β2s were not significantly different among the 3 clusters (S5 Appendix).

## Interaction of percent weight change and PHQ-9 score clusters

The cross-classification of participants according to the clusters of percent weight change and PHQ-9 scores and participants with insufficient data (cluster 0) is shown in Table 2. Three

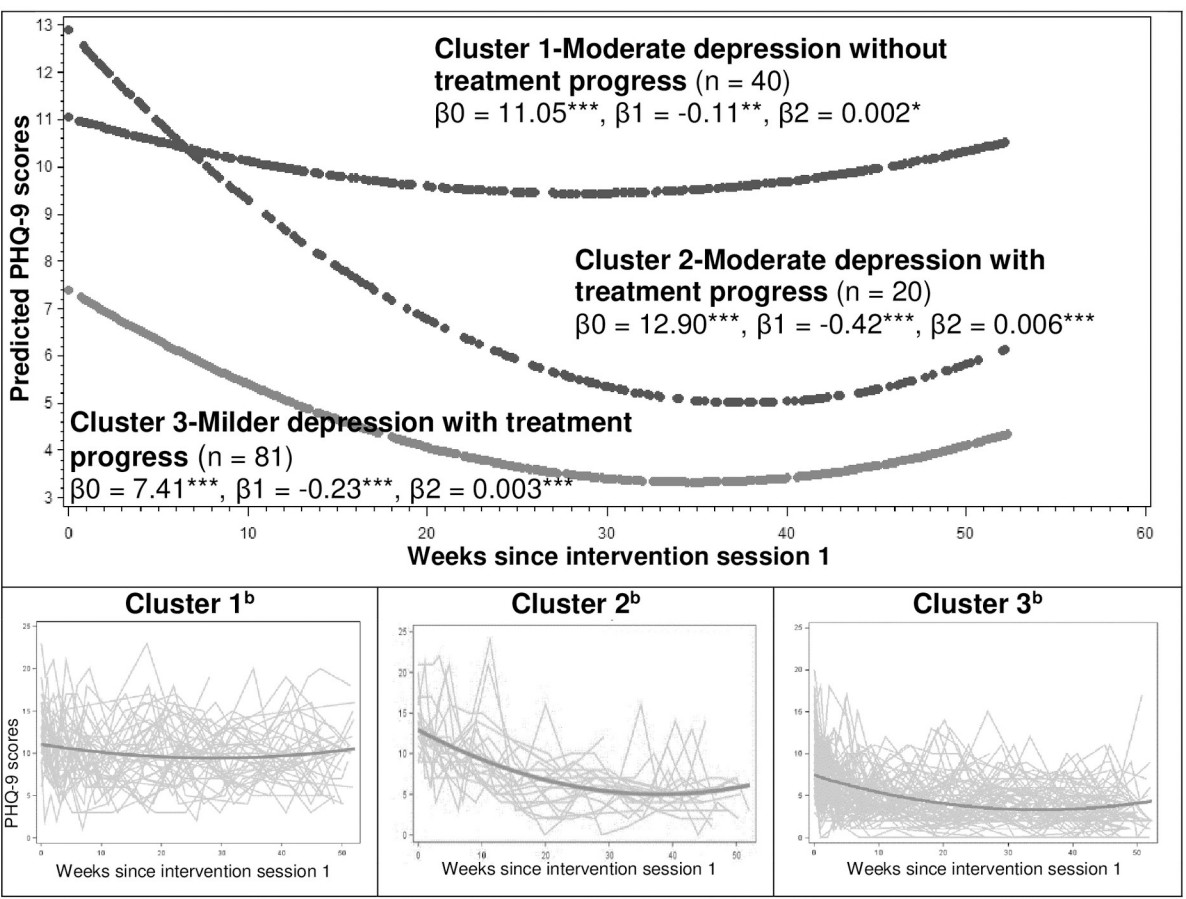

β0 = intercept; β1 = linear coefficient; β2 = quadratic coefficient. *P<.05; **P<.01; ***P<.001
[a]141 participants, or 69% of the intervention group (n=204), had PHQ-9 data in at least 3
quarters of the 12-month intervention period.
[b]Light gray lines show individual participant trajectories within each cluster.

**Fig 2. PHQ-9 trajectories among intervention participants with PHQ-9 data in at least 3 quarters of the 12-month intervention period[a,b].** β0 = intercept; β1 = linear coefficient; β2 = quadratic coefficient. *P < .05; **P < .01; ***P < .001. [a]141 participants, or 69% of the intervention group (n = 204), had PHQ-9 data in at least 3 quarters of the 12-month intervention period. [b]Light gray lines show individual participant trajectories within each cluster.

categories were identified. The poor treatment engagement (n = 63) had both poor session attendance and all but 2 participants had poor self-monitoring of weight and consequently inadequate data to be included in cluster analysis. The poor treatment progress (n = 80) had minimal improvement in self-monitored weight or PHQ-9 or had poor self-monitoring of weight despite attending sessions. The progress category (n = 61) showed overall positive treatment progress for both self-monitored weight and PHQ-9. There were minimal differences in baseline characteristics among these categories (S6 Appendix).

### Validation of treatment engagement and progress categories

These categories differed significantly in the indices of behavioral adherence to the intervention and objectively measured weight loss and depression outcomes at 6 and 12 months (Table 3). Participants with poor engagement attended fewer than 5 out of 15 sessions (SD 2.6) and provided self-monitored data in a minimal number of weeks either actively (manual

**Table 2. Categories of treatment engagement and progress based on percent weight change and PHQ-9 clusters.**

| | Frequency Percent Row Percent Column Percent | PHQ-9 trajectory cluster | | | | Total |
|---|---|---|---|---|---|---|
| | | 0-No PHQ-9 cluster (i.e., Poor session attendance) (n = 63) | 1-Moderate depression without treatment progress (n = 40) | 2-Moderate depression with treatment progress (n = 20) | 3-Milder depression with treatment progress (n = 81) | |
| Percent weight change trajectory cluster | 0-No weight cluster (i.e., Poor self-monitoring) (n = 81) | 61 | 11 | 1 | 8 | 81 |
| | | 29.9 | 5.4 | 0.5 | 3.9 | 39.7 |
| | | 75.3 | 13.6 | 1.2 | 9.9 | |
| | | 96.8 | 27.5 | 5.0 | 9.9 | |
| | 1-Minimal weight loss (n = 50) | 1 | 18 | 9 | 22 | 50 |
| | | 0.5 | 8.8 | 4.4 | 10.8 | 24.5 |
| | | 2.0 | 36.0 | 18.0 | 44.0 | |
| | | 1.6 | 45.0 | 45.0 | 27.2 | |
| | 2-Moderate weight loss (n = 61) | 1 | 10 | 8 | 42 | 61 |
| | | 0.5 | 4.9 | 3.9 | 20.6 | 29.9 |
| | | 1.6 | 16.4 | 13.1 | 68.9 | |
| | | 1.6 | 25.0 | 40.0 | 51.9 | |
| | 3-Most weight loss (n = 12) | 0 | 1 | 2 | 9 | 12 |
| | | 0.0 | 0.5 | 1.0 | 4.4 | 5.9 |
| | | 0.0 | 8.3 | 16.7 | 75.0 | |
| | | 0.0 | 2.5 | 10.0 | 11.1 | |
| | Total | 63 | 40 | 20 | 81 | 204 |
| | | 30.9 | 19.6 | 9.8 | 39.7 | 100.0 |

Different shades indicate the 3 categories of treatment engagement and progress: (1) light gray: the poor engagement category (n = 63), (2) gray: the poor progress category (n = 80), and (3) dark gray: the progress category (n = 61).

entries of weight or minutes of leisure-time physical activity in <3 weeks) or passively (FitBit steps uploaded automatically in <9 weeks). Among them, 39 (62%) discontinued the intervention by session 5 (week 6) and another 11 (17%) dropped out at session 6 (week 8). Relatedly, these participants also had minimal improvements in both weight loss and depression outcomes at 6 and 12 months. Participants with poor treatment progress attended almost all 15 sessions and showed intermediate levels of self-monitoring. However, the objectively measured weight loss and depression outcomes among these participants were comparable to participants with poor engagement and worse than participants with treatment progress. The last category had perfect attendance and good active (weight or physical activity minutes monitored for 21 to 22 weeks) and passive self-monitoring (FitBit steps uploaded in 36 weeks). Mean (SD) reductions were -2.0 (1.3) in BMI and -0.6 (0.6) in SCL-20 at 6 months, which sustained at 12 months. These categories also differed significantly in the number of days since Session 1 for each subsequent in-person session, possibly reflecting different degrees of scheduling difficulties, disinterest, or lack of commitment (S7 Appendix).

## Sensitivity analysis

Among participants with both self-monitored weight and PHQ-9 score in all 4 quarters (n = 88), joint cluster analysis resulted in 3 clusters: (A) no treatment progress in either percent weight change or PHQ-9 (n = 19); (B) treatment progress in PHQ-9 only (n = 33); and (C) treatment progress in both (n = 36) (S8 Appendix). Separate cluster analysis resulted in 2 clusters for percentage weight change: without (n = 48) and with (n = 40) weight loss; and 2 clusters for PHQ-9: without (n = 26) and with (n = 62) treatment progress (S8 Appendix). The

**Table 3. Comparisons of adherence behaviors and outcomes by category of treatment engagement and progress.**

| | All intervention participants (n = 204) | Poor engagement (n = 63; 31%) | Poor progress (n = 80; 39%) | Progress (n = 61; 30%) | P value |
|---|---|---|---|---|---|
| **Adherence behaviors** | | | | | |
| No. of sessions attended | 11.4 (5.1) | 4.5 (2.6)[a] | 14.2 (2.3)[b] | 15.0 (0.1)[c] | <0.001 |
| No. of weeks with self-monitored weight | 11.5 (11.5) | 2.6 (4.3)[a] | 11.2 (8.5)[b] | 21.0 (12.6)[c] | <0.001 |
| No. of weeks with self-reported minutes of leisure-time physical activity | 11.5 (12.0) | 1.5 (2.3)[a] | 11.6 (8.5)[b] | 21.8 (13.2)[c] | <0.001 |
| No. of weeks with FitBit steps | 24.5 (17.4) | 8.7 (9.2)[a] | 28.0 (15.8)[b] | 36.1 (13.8)[c] | <0.001 |
| **Weight loss and depression outcomes** | | | | | |
| BMI change from baseline | | | | | |
| 6 months, n = 196 | -0.7 (1.7) | 0.0 (1.8)[a] | -0.2 (1.4)[a] | -2.0 (1.3)[b] | <0.001 |
| 12 months, n = 183 | -0.7 (2.2) | 0.1 (2.4)[a] | -0.0 (1.6)[a] | -2.2 (1.9)[b] | <0.001 |
| ≥5% weight loss from baseline, No. (%) | | | | | |
| 6 months, n = 196 | 48 (24.5) | 8 (14.0)[a] | 8 (10.1)[a] | 32 (53.3)[b] | <0.001 |
| 12 months, n = 183 | 51 (27.9) | 8 (16.7)[a] | 10 (13.2)[a] | 33 (55.9)[b] | <0.001 |
| SCL-20 change from baseline | | | | | |
| 6 months, n = 175 | -0.3 (0.7) | 0.0 (0.7)[a] | -0.3 (0.7)[b] | -0.6 (0.6)[b] | <0.001 |
| 12 months, n = 169 | -0.3 (0.7) | 0.3 (0.8)[a] | -0.3 (0.7)[b] | -0.6 (0.6)[b] | <0.001 |
| Depression response (≥50% decrease in SCL-20 scores from baseline), No. (%) | | | | | |
| 6 months, n = 175 | 55 (31.4) | 5 (13.5)[a] | 19 (24.4)[a] | 31 (51.7)[b] | <0.001 |
| 12 months, n = 169 | 49 (29.0) | 4 (12.1)[a] | 18 (23.4)[a] | 27 (45.8)[b] | 0.001 |
| Depression remission (SCL-20 scores < 0.5), No. (%) | | | | | |
| 6 months, n = 175 | 31 (17.7) | 1 (2.7)[a] | 11 (14.1)[a] | 19 (31.7)[b] | <0.001 |
| 12 months, n = 169 | 30 (17.8) | 3 (9.1)[a] | 10 (13.0)[a] | 17 (28.8)[b] | 0.02 |

Values are mean (SD) unless otherwise noted. P values are obtained from ANOVA comparing 3 categories for continuous variables or from the chi-square test comparing 3 categories for categorical variables.

[a, b, c] Different superscripts denote statistically significant differences between categories.

number of participants in clusters resulting from joint and separate cluster analyses showed high concordance. For example, 52 participants with minimal weight loss (i.e., clusters A and B) in the joint cluster analysis compared to 48 (cluster 1) in the cluster analysis on weight only. Also, 69 participants with PHQ-9 response (i.e., clusters B and C) in the joint cluster analysis compared to 62 (cluster 2) in the cluster analysis on PHQ-9 only. In addition, the number of participants in clusters resulting from the joint cluster analysis also showed concordance with the number of participants in Table 2. For example, the number of 33 participants with treatment progress in PHQ-9 only resulting from the joint cluster analysis (i.e., group B in S8 Appendix) was concordant with the number of 31 participants who had depression treatment progress (i.e., PHQ-9 cluster 2 and 3) but minimal weight loss (i.e., weight change cluster 1) in Table 2.

## Discussion

This study showed that even in the context of an efficacious intervention for obesity and depression, participants varied in treatment engagement and progress. Poor treatment engagement manifested as low adherence to session attendance and/or self-monitoring affected >30% of participants. Among those engaged patterns of treatment progress differentiated for weight loss: (1) minimal weight loss, (2) moderate weight loss, and (3) substantial weight loss;

and for depression: (1) initial moderate depressive symptoms without progress, (2) initial moderate depressive symptoms with progress, and (3) initial milder depressive symptoms with progress. These patterns were not only significantly associated with intervention adherence behaviors—such as session attendance and self-monitoring—but also with objectively assessed efficacy outcomes.

Evidence on this topic is scarce due to limited data assessment points in conventional clinical trials. Only a few prior studies have investigated the dynamic trajectories of weight loss in behavioral interventions and, to a lesser extent, the trajectories of depression symptoms. The 3 weight loss patterns identified in the current study were similar to those found in previous weight loss studies. [34, 35] The current study also identified 3 depression symptom patterns, similar to—although not identical to—the 2 prior studies investigating dynamic trajectories of depression symptoms. These prior studies identified 2 patterns, gradual/slower responders and rapid responders (in an antidepressant only or antidepressant plus psychotherapy study). [36–38] one study [36] found that higher baseline depression severity was associated with the gradual/slower responder trajectory; whereas the current study found that among participants with higher baseline depression severity there were 2 distinct subgroups, those with treatment progress and those without, although this sample had comorbid obesity.

To our knowledge, the present study is the first to examine the temporal patterns of change in both weight and depression severity in response to an integrated collaborative care intervention for comorbid obesity and depression. Separation of the weight loss and PHQ-9 clusters was evident by 6 to 8 weeks of treatment and persisted throughout the 12-month intervention. Additionally, this study identified subgroups of treatment engagement and progress levels that were significantly associated with objectively-measured weight loss and depression outcomes at the end of the intervention. These findings suggest that evaluation of dynamic treatment engagement and progress early in the course of intervention might provide important information regarding how an individual will respond by the intervention endpoint. Consistent with this study, earlier studies on weight loss demonstrated that initial weight loss at 1 to 2 months was significantly associated with 1-year and even longer-term weight loss up to 8 years. [39–41] This study found that participants with poor engagement or poor progress showed minimal differences in baseline characteristics from those with progress; however, they differed significantly in intervention adherence behaviors. This has practical implications because poor adherence behaviors—such as low rates of session attendance and self-monitoring—are easy to detect and respond to early in the course of an intervention. Treatment strategies could be adjusted for these individuals to optimize treatment outcomes. For example, participants who show early signs of nonengagement such as poor session attendance and/or self-monitoring or poor progress such as not reaching interim intervention goals may benefit from an augmented intervention with motivational interviewing strategies, thereby minimizing the risk of treatment failure. Similarly, a recent study of a community-based intervention for chronic disease management in participants with two or more diseases (i.e., diabetes, obesity, hypertension, and tobacco dependence) reported that treatment response was predicted by participants' reactions to the challenges and failures they faced during the intervention instead of their baseline characteristics; the authors concluded that behavioral interventions could be modified to help non-responders face the challenges and failures. [42]

This study has limitations. First, because of the post hoc nature of the analyses the findings need to be replicated in future studies. Second, the weight change and PHQ-9 score clusters may be specific to the study data. Therefore, future studies of independent samples are needed to verify the external validity of the results. In addition, given that participants might be less severely depressed in RCTs, [43] this may reduce generalizability of the findings to the clinical population with more severe depression. Third, this study only evaluated trajectories of weight

change and depression symptoms over 12 months; consequently, it does not provide insight into trajectories of long-term outcomes.

## Conclusions

This study carefully examined heterogeneity in treatment engagement and progress over the course of an efficacious, yearlong integrated collaborative care intervention for obesity and depression and identified subgroups of patients who were more or less likely to engage in or benefit from this type of treatment. Signs of poor engagement or progress manifested early in the intervention, persisted, and correlated significantly with treatment efficacy outcomes. Identifying patients with likely treatment failure using engagement and progress data early in the intervention could enable individualized optimization to enhance efficacy.

## Supporting information

**S1 Appendix. Intervention outline[a,b].** a In-between session support as needed via EHR secure email, between weeks 1–52. b Co-located psychiatric and medical supervision during weekly intervention management team meeting, between weeks 1–52. c The 9 one-on-one I-CARE sessions will occur primarily in the clinic, but video conferences (as the second option) or phone sessions (as the last option and for visits 1–5, phone session is only an option upon PI and intervention manager approval) throughout the intensive phase will be an option for participants with considerable constraints. d I-CARE Mood is the PEARLS program; I-CARE Lifestyle is the GLB program. e Participants receive Fitbit, MyFitnessPal, and My Health Online instructions via mail or e-mail prior to first session.
(DOCX)

**S2 Appendix. Individual participant trajectories within each cluster of percent weight change.**
(DOCX)

**S3 Appendix. Mean (±SD) beta coefficients of individual trajectories within each cluster of percent weight change.** $\beta1$ = Linear coefficient; $\beta2$ = Quadratic coefficient. [abc]Different letters indicate significant difference.
(DOCX)

**S4 Appendix. Individual participant trajectories within each cluster of PHQ-9 change.**
(DOCX)

**S5 Appendix. Mean (±SD) beta coefficients of individual trajectories within each cluster of PHQ-9 change.** $\beta0$ = Intercept; $\beta1$ = Linear coefficient; $\beta2$ = Quadratic coefficient. [abc]Different letters indicate significant difference.
(DOCX)

**S6 Appendix. Comparisons of baseline characteristics by category of treatment engagement and progress.** Abbreviations: BMI, body mass index; PHQ-9, Patient Health Questionnaire-9; SCL20, Symptom Checklist-20. Values are mean (SD) unless otherwise noted.
(DOCX)

**S7 Appendix. Mean (±SD) number of days from session 1 by category of treatment engagement and progress.** Abbreviations: NA, not applicable (SD is NA because only 1 participant attended session 9 in the poor engagement category). [a, b, c] Different superscripts denote statistically significant differences between categories.
(DOCX)

**S8 Appendix. Joint and separate cluster analysis of weight and PHQ-9 trajectories among intervention participants who had at least one self-monitored weight measure and one PHQ-9 score in all 4 quarters (n = 88).**
(DOCX)

## Acknowledgments

The following research team members contributed instrumentally to the delivery of the I-CARE intervention: Andrea Blonstein, MBA, RD (Sutter Health) and Hoang Nguyen (Blue Shield of California).

Dr. Ma had full access to all the data in the study and takes responsibility for the integrity of the data and the accuracy of the data analysis.

Concept and design: Ma, Lavori.

Acquisition, analysis, or interpretation of data: All authors.

Drafting of the manuscript: Lv, Xiao, Majd.

Critical revision of the manuscript for important intellectual content: All authors.

Statistical analysis: Lv, Xiao, Lavori, Ma.

Obtaining funding: Ma, Lewis, Lavori.

Administrative, technical, or material support: Lv, Rosas, Azar, Snowden, Venditti, Lewis, Ward, Lesser, Ma.

Supervision: Ma, Rosas.

## Author Contributions

**Conceptualization:** Nan Lv, Lan Xiao, Philip W. Lavori, Joshua M. Smyth, Lisa G. Rosas, Elizabeth M. Venditti, Mark B. Snowden, Megan A. Lewis, Leanne M. Williams, Jun Ma.

**Formal analysis:** Nan Lv, Lan Xiao, Philip W. Lavori.

**Funding acquisition:** Philip W. Lavori, Megan A. Lewis, Jun Ma.

**Investigation:** Philip W. Lavori, Jun Ma.

**Methodology:** Lan Xiao, Philip W. Lavori, Jun Ma.

**Project administration:** Nan Lv, Lisa G. Rosas, Elizabeth M. Venditti, Mark B. Snowden, Megan A. Lewis, Elizabeth Ward, Lenard Lesser, Kristen M. J. Azar.

**Supervision:** Philip W. Lavori, Lisa G. Rosas, Jun Ma.

**Validation:** Nan Lv, Lan Xiao.

**Visualization:** Nan Lv, Lan Xiao.

**Writing – original draft:** Nan Lv, Lan Xiao, Marzieh Majd, Jun Ma.

**Writing – review & editing:** Nan Lv, Lan Xiao, Marzieh Majd, Philip W. Lavori, Joshua M. Smyth, Lisa G. Rosas, Elizabeth M. Venditti, Mark B. Snowden, Megan A. Lewis, Elizabeth Ward, Lenard Lesser, Leanne M. Williams, Kristen M. J. Azar, Jun Ma.

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
