## [Decision Letter · Decision Letter 0]

18 Sep 2019

PONE-D-19-15164

Variability in engagement and progress in efficacious integrated collaborative care for primary care patients with obesity and depression: within-treatment analysis in the RAINBOW trial

PLOS ONE

Dear Dr. Ma,

Thank you for submitting your manuscript to PLOS ONE. After careful consideration, we feel that it has merit but does not fully meet PLOS ONE’s publication criteria as it currently stands. Therefore, we invite you to submit a revised version of the manuscript that addresses the points raised during the review process.

The study is timely. However, one reviewer has recommended a rejection. In my opinion, the data are interesting, and manuscript can be published if all questions/comments are addressed and presentation is revised. Presently it is impossible to understand for an “outsider”. 

Overall, it appears that on the one hand it is understandable for any physician that a patient who is not taking medications, i.e. not coming to sessions or does not do required assignment (checking weight), i.e. “not engaged” is usually not expected to improve in any disease for any outcome. So, a priori cluster 0, n=61 is not expected to improve and it appears that this is your conclusion: “Participants with failed treatment outcomes demonstrated poor engagement or progress early in the intervention. This insight could inform individualized optimization to enhance efficacy.” Perhaps it can be acceptable if other components of the paper are improved as requested by the reviewers. If this is your major conclusion you have to incorporate this “intuitive” thinking somewhere. It is obvious but still valuable since published papers do not commonly talk about this.

On the other hand, Table 3 shows that if you compare “Weight % change” in “poor progress” and “Progress” there is significant weight change differences. The Table shows that both groups attended practically the same number of sessions, n=14 and n=15, respectively (yet statistically significant?). Also, although self-monitoring (for weight?) appears different 11.2 vs. 21 weeks, it is not statistically different. In addition, for “weight % change”, n=196 or n=183. What does this mean? Analysis included 123 participants for weight according to Methods. All this is confusing. Each of these points has to be addressed.

There are multiple “small” inconsistencies that will need to be addressed after major problems are resolved. For example: 

1. the definition of “engagement” only appears in Results, when it is expected in Methods.

2. Table 1 shows 204 participants. However, this sub-analysis includes only 123 or 141 participants, so data for these are expected.

3. Is this “post hoc” analysis of previous data? Was “Engagement” assessed after study was completed?

4. Did you try to answer a question: Who failed the treatment?

5. Use n=123 and n=141 in all analyses and in Tables.

6. Since weight loss and depression are assessed separately, present the outcomes separately to simplify understanding. Use n=141 for all data on depression and n=123 for all data on weight loss. Then address interaction between these outcomes.

We would appreciate receiving your revised manuscript by Nov 02 2019 11:59PM. To enhance the reproducibility of your results, we recommend that if applicable you deposit your laboratory protocols in protocols.io, where a protocol can be assigned its own identifier (DOI) such that it can be cited independently in the future. For instructions see: http://journals.plos.org/plosone/s/submission-guidelines#loc-laboratory-protocols

We look forward to receiving your revised manuscript.

Kind regards,

Elena Barengolts, MD

Academic Editor

PLOS ONE

Journal Requirements:

3. Please ensure you have included the registration number for the clinical trial referenced in the manuscript

'The authors have declared that no competing interests exist.' 

We note that one or more of the authors are employed by a commercial company: One Medical.

Additional Editor Comments (if provided):

Reviewers' comments:

Reviewer's Responses to Questions

**Comments to the Author**

1. Is the manuscript technically sound, and do the data support the conclusions?

Reviewer #1: Partly

Reviewer #2: Yes

2. Has the statistical analysis been performed appropriately and rigorously? 

Reviewer #1: No

Reviewer #2: Yes

3. Have the authors made all data underlying the findings in their manuscript fully available?

Reviewer #1: No

Reviewer #2: Yes

4. Is the manuscript presented in an intelligible fashion and written in standard English?

Reviewer #1: Yes

Reviewer #2: Yes

5. Review Comments to the Author

Reviewer #1: ABSTRACT

I do not understand the regression weights in the abstract. I also do not understand how these are related to outcome. Rather give effect sizes that are interpretable and clearly describe the variables please. Furthermore, if you want to show differences between groups in efficacy - what I think that you want to shoe, you should present interaction effects toon differences between sub-groups (and not main effects in sub-groups).

METHOD

I also do not understand the method sections and hence also not the results. Perhaps its me I don’t know. But based on what I read and what I understand from it, I do not recommend publishing this paper

Reviewer #2: This study investigates the combined variation in body weight changes and depression scores among patients undergoing an integrated care programme. This is a very worthwhile objective, in order to describe and understand the spectrum of reality behind mean and standard deviation summary values. The authors successfully describe the variation by means of cluster analysis. However, although attempts were made to verify the results using sensitivity analyses, it remains unclear to what extent the clusters were distinct and whether the chosen ‚optimal‘ number of clusters is a characteristic of the data or a construct of the analysis method.

Further, the conclusion line in the abstract ‘Participants with failed treatment outcomes demonstrated poor engagement or progress early in the intervention’ does not, to me, seem to be clearly demonstrated. Most of those with poor engagement had no data on either weight loss or depression. And particularly for depression scores, many of those in cluster 1 never-the-less had improvements in the early period. The authors need to argue this point more exactly.

Specific points:

1. The sensitivity analysis based jointly on weight and PHQ scores is particularly interesting since it yields a cluster whose members responded in PHQ but not in weight. This result is not evident (or at least, not emphasized) in the main cluster analysis. The concordance of this clustering result with the main analysis and the display in Table 3 might be informative.

2. I suggest using, as a sensitivity analysis, clustering based not on the quarterly means but on an alternative averaging over time (e.g. 4-monthly). Further, an alternative clustering method without pre-specification of the number of clusters might help to test the robustness of the results.

3. Figs. 1 and 2 display the individual trajectories but they are so small that it is difficult to get the picture. Could larger plots be shown?

4. The regression coefficient values given in appendix tables S2 and S3 differ from those in the text (pp. 9 and 10).

5. Many the ‘observed’ correlations and ‘significant’ differences are tautologous, stemming from the same or related data. For instance, the clusters were defined based on the weight and depression score changes, so it is almost inevitable that the clusters differ significantly with regard to the regression coefficients for these changes. It is similarly to be expected that the categories of treatment engagement and progress differ significantly with respect to adherence behaviours, since these behaviours were used to define the ‚no weight‘ and ‚no PHQ-9‘ clusters and thus the first category.

6. Did the authors test whether the polynomial models of trajectories fitted the data well, either to the individual trajectories or to the cluster means?

7. It would be interesting to investigate the possible chronological and causal relationships between weight change and depression score change: were they simultaneous or did one tend to follow the other? Or maybe no regularity can be discerned?

6. PLOS authors have the option to publish the peer review history of their article (what does this mean?). If published, this will include your full peer review and any attached files.

Reviewer #1: Yes: Marc Molendijk

Reviewer #2: Yes: Jeremy Franklin

---

## [Author Response · Author response to Decision Letter 0]

19 Nov 2019

We thank the editor and reviewers for their constructive comments on our originally submitted manuscript, titled “Variability in engagement and progress in efficacious integrated collaborative care for primary care patients with obesity and depression: within-treatment analysis in the RAINBOW trial.” Below please find our point-to-point responses.

Editor’s comment:

1. Overall, it appears that on the one hand it is understandable for any physician that a patient who is not taking medications, i.e. not coming to sessions or does not do required assignment (checking weight), i.e. “not engaged” is usually not expected to improve in any disease for any outcome. So, a priori cluster 0, n=61 is not expected to improve and it appears that this is your conclusion: “Participants with failed treatment outcomes demonstrated poor engagement or progress early in the intervention. This insight could inform individualized optimization to enhance efficacy.” Perhaps it can be acceptable if other components of the paper are improved as requested by the reviewers. If this is your major conclusion you have to incorporate this “intuitive” thinking somewhere. It is obvious but still valuable since published papers do not commonly talk about this.

Authors’ response: We agree with the intuitiveness of our finding that patients with poor treatment engagement had poor outcomes. In addition to stating this obvious finding (which we also agree is nonetheless valuable), we underscored its potential clinical implications in the third paragraph of the discussion section: The percent weight change and PHQ-9 clusters diverged at the beginning of the intervention, and the clusters correlated significantly with adherence behaviors that are easier to monitor early in an intervention in the real world. These findings suggest that the adherence behaviors and early signs of poor progress (within weeks) based on self-monitored weight and PHQ-9 scores could be used to identify participants who would not achieve clinically significant improvements in BMI or SCL-20 at the end of the 12-month treatment. In doing so, alternative strategies may be provided to augment the treatment in order to help people move from the poor engagement or poor progress group to the progress group. As detailed below, we responded to all the editor’s and reviewers’ other comments and suggestions, and we believe that the revised manuscript has improved substantially as a result.

2. On the other hand, Table 3 shows that if you compare “Weight % change” in “poor progress” and “Progress” there is significant weight change differences. The Table shows that both groups attended practically the same number of sessions, n=14 and n=15, respectively (yet statistically significant?). Also, although self-monitoring (for weight?) appears different 11.2 vs. 21 weeks, it is not statistically different. In addition, for “weight % change”, n=196 or n=183. What does this mean? Analysis included 123 participants for weight according to Methods. All this is confusing. Each of these points has to be addressed.

Authors’ response: Owing to formatting issues, some numbers in Table 3 were truncated; this problem is now fixed. As indicated in Table 3, the P values were obtained from ANOVA or chi-square tests comparing 3 groups, “poor engagement,” “poor progress,” and “progress” while significant pairwise differences were denoted by superscripts. The different numbers of participants with weight data (n=123, 196, 183) were correct and were attributed to differences in the sources of weight measurements. Weight data used in the cluster analysis were self-monitored weights provided by participants throughout the intervention, and 123 was the number of participants included in the cluster analysis for weight change over time because they met the criterion of having self-monitored weight data in at least 3 quarters during the one-year intervention. This criterion was applied to enhance the reliability of weight change patterns and reduce the influence of participants with missing data in 2 or more quarters. Weight data used to validate the results from cluster analyses as reported in Table 3 were weights objectively measured at baseline, 6, and 12 months by research staff blinded to treatment assignment. Of the 204 intervention participants, the number of participants with objectively-measured weights was 196 at 6 months and 183 at 12 months. We have clarified the different sources of weight data in the measures section (see pages 5-7 lines 128-161). 

3. The definition of “engagement” only appears in Results, when it is expected in Methods.

Authors’ response: We have moved the definitions of the 3 categories of treatment engagement and progress to the methods section: “The poor engagement category included participants who had poor session attendance (i.e., cluster 0 for PHQ-9). The poor progress category included participants who had minimal improvement in self-monitored weight or PHQ-9 (i.e., cluster 1 for either) or had poor self-monitoring of weight despite attending sessions (i.e., cluster 0 for weight and cluster 1, 2, or 3 for PHQ-9). The progress category included participants who had improvements in both self-monitored weight and PHQ-9 (i.e., cluster 2 or 3 for both)” (see page 9 lines 208-214).

4. Table 1 shows 204 participants. However, this sub-analysis includes only 123 or 141 participants, so data for these are expected.

Authors’ response: The total number of participants in the intervention was 204. In Table 1 and the results section, we have added baseline characteristics of the subgroups of participants with data in at least 3 quarters during the one-year intervention to be included in cluster analyses of percent weight change (n=123) and PHQ-9 scores (n=141).

5. Is this “post hoc” analysis of previous data? Was “Engagement” assessed after study was completed?

Authors’ response: Yes, this is a “post hoc” analysis of the data collected in the RAINBOW trial. We have clarified this in the methods of the abstract (page 2 line 38) and in the last paragraph of the introduction (page 4 line 90). In addition, we acknowledge the post hoc nature of the study as a limitation in the discussion, which was in the original submission as well. It is also correct that engagement was assessed after the study was completed. In the statistical analysis section under the “categorization and validation of treatment engagement and progress” subheading, we defined the 3 categories of treatment engagement and progress (i.e., poor engagement, poor progress, and progress) (page 9 lines 208-214). 

6. Did you try to answer a question: Who failed the treatment?

Authors’ response: Our analyses identifying participants with poor engagement or poor progress address the question “Who failed the treatment?” We have added in the results (page 13 lines 297-298) with a new Appendix table (S6) to clarify that there were minimal differences in baseline characteristics among the 3 categories (poor engagement, poor progress, and progress). However, as reported in our original submission, these categories differed significantly in multiple indices of behavioral adherence to the intervention (Table 3). In the discussion, we have added the following with citation of a recent study (Ed et al., 2018): “This study found that participants with poor engagement or poor progress showed minimal differences in baseline characteristics from those with progress; however, they differed significantly in intervention adherence behaviors. …… Similarly, a recent study of a community-based intervention for chronic disease management in participants with two or more diseases (i.e., diabetes, obesity, hypertension, and tobacco dependence) reported that treatment response was predicted by participants’ reactions to the challenges and failures they faced during the intervention instead of their baseline characteristics; the authors concluded that behavioral interventions could be modified to help non-responders face the challenges and failures.” 

7. Use n=123 and n=141 in all analyses and in Tables.

Authors’ response: By responding to comment #4 above, we have added n=123 and n=141 in Table 1 to show demographics for these two subgroups. The subgroup of 123 participants included those who met the criteria to be included in cluster analysis for percent weight change. The subgroup of 141 participants included those who met the criteria to be included in cluster analysis for PHQ-9 scores. In Table 2, we have added the numbers of participants in cluster 0, 1, 2, and 3 for percent weight change and PHQ-9. The sum of clusters 1, 2, and 3 for percent weight change was 123, and the sum of clusters 1, 2, and 3 for PHQ-9 was 141. Cluster 0 included participants who did not have adequate data to be included in the cluster analyses. The four clusters combined capture all 204 intervention participants. Table 3 shows adherence behaviors and objectively measured weight and depression outcomes by category of treatment engagement and progress, which was created based on combinations of the four percent weight change clusters and the four PHQ-9 clusters as displayed in Table 2. Therefore, it is impossible to identify n=123 and n=141 in Table 3.

8. Since weight loss and depression are assessed separately, present the outcomes separately to simplify understanding. Use n=141 for all data on depression and n=123 for all data on weight loss. Then address interaction between these outcomes.

Authors’ response: We indeed presented the results as recommended. Figures 1 and 2 present cluster analysis results for the trajectories of percent weight change and PHQ-9 scores separately. Then, Tables 2 and 3 present results combining the clusters for each outcome to examine their interactions. To further clarify this order of presentation, we have added subheadings in the results section.

Authors’ response: Done.

2. We note that you have indicated that data from this study are available upon request. PLOS only allows data to be available upon request if there are legal or ethical restrictions on sharing data publicly. 

Authors’ response: We have provided the information in the cover letter: “The IRB-approved study protocol and written informed consent forms provided by participants in the study do not permit sharing of de-identified data without a formal data sharing and use agreement. For data requests, the University of Illinois at Chicago (UIC) IRB can be reached at 1-(312) 996-1711 or uicirb@uic.edu.”

Authors’ response: Please see our response above.

3. Please ensure you have included the registration number for the clinical trial referenced in the manuscript

Authors’ response: We did include the registration number (ClinicalTrials.gov#NCT02246413) below abstract in the original submission and have kept it there in this resubmission. 

'The authors have declared that no competing interests exist.' 

We note that one or more of the authors are employed by a commercial company: One Medical.

Authors’ response: The NIH was the sole funder of the reported research, and there was no commercial affiliation. Our Finding Statement in the cover letter is as follows: “Research reported in this publication was supported by the National Heart, Lung, And Blood Institute of the National Institutes of Health (URL: https://www.nhlbi.nih.gov/) under Award Number R01HL119453 (Recipient: JM). The funder had no role in study design, data collection and analysis, decision to publish, or preparation of the manuscript. The content is solely the responsibility of the authors and does not necessarily represent the official views of the National Institutes of Health. There were no other funding sources.”

Authors’ response: It is accurate that the authors had no competing interests. Our Competing Interests Statement in the cover letter is as follows: “The authors declare that no competing interests existed for the research as reported. Dr. Lenard Lesser’s affiliation with 1Life Healthcare/One Medical constituted no competing interests. Dr. Lesser had originally supported on the NIH grant for his role as a study physician on this study while he was an employee at the Palo Alto Medical Foundation Research Institute (PAMFRI) where the study was conducted. Starting from 07/2016 till 10/2017, Dr. Lesser transitioned from PAMFRI to 1Life Healthcare, Inc/One Medical. At that time, his continuous involvement in the study was compensated through a research contract as an independent consultant with PAMFRI specifically for the NIH grant supporting the study. 1Life Healthcare/One Medical provided no support of any form for the study; and had no role in the study design, data collection/analysis, decision to publish, or preparation of the manuscript. This does not alter our adherence to PLOS ONE policies on sharing data and materials.”

Reviewer 1’s comments:

ABSTRACT

I do not understand the regression weights in the abstract. I also do not understand how these are related to outcome. Rather give effect sizes that are interpretable and clearly describe the variables please. Furthermore, if you want to show differences between groups in efficacy - what I think that you want to show, you should present interaction effects on differences between sub-groups (and not main effects in sub-groups).

Authors’ response: The objective of this paper was not to show differences in efficacy outcomes between the intervention and control groups, which has been published in JAMA (Ma et al., JAMA 2019, 321(9):869-879). Instead, this paper aimed to examine different patterns of engagement and progress and the relation of these patterns to outcomes among participants within the intervention group only. In other words, the objective of this paper was to understand participants’ response or lack thereof to the intervention so that future research can be designed to address non-responders and enhance intervention efficacy. The introduction of the abstract states, “The RAINBOW randomized clinical trial validated the efficacy of an integrated collaborative care intervention for obesity and depression in primary care, although the effect was modest. To inform intervention optimization, this study investigated within-treatment variability in participant engagement and progress.” Beta coefficients are the appropriate effect estimates for the cluster analyses as reported showing patterns of weight change (%) and PHQ-9 scores over the 12-month intervention. As noted in the abstract, β0 is the intercept, β1 is the linear term, and β2 is the quadratic term. 

METHOD

I also do not understand the method sections and hence also not the results. Perhaps its me I don’t know. But based on what I read and what I understand from it, I do not recommend publishing this paper.

Authors’ response: The reviewer indicated lack of understanding of the methods and results sections without specific comments that we could effectively address. We appreciate the opportunity to revise and resubmit our manuscript per the editor’s decision. We agree with the editor that the study has merit and is timely. We have indeed revised our methods and results sections by responding to the editor’s and reviewer 2’s comments, and we believe the revised manuscript is substantially improved as a result. We welcome any additional specific feedback from the editor and reviewers to further improve the manuscript for publication.

Reviewer 2’s comments: 

1. This study investigates the combined variation in body weight changes and depression scores among patients undergoing an integrated care program. This is a very worthwhile objective, in order to describe and understand the spectrum of reality behind mean and standard deviation summary values. The authors successfully describe the variation by means of cluster analysis. However, although attempts were made to verify the results using sensitivity analyses, it remains unclear to what extent the clusters were distinct and whether the chosen ‘optimal’ number of clusters is a characteristic of the data or a construct of the analysis method.

Authors’ response: In the original manuscript, we described the separation of these clusters and included detailed results in Appendices. In the revised manuscript, we have further clarified the results on clusters of percent weight change: “Pairwise comparisons of β1 and β2 coefficients of the individual trajectories across the 3 clusters showed that these clusters were distinct from each other: β1 of cluster 3 was significantly lower than cluster 2, which was significantly lower than cluster 1; and β2 of cluster 3 was not significantly different from cluster 2 but both were significantly higher than cluster 1 (S3 Appendix)” (see page 11 lines 244-247). We have also further clarified the results on clusters of PHQ-9 scores: “pairwise comparisons of β0, β1, and β2 coefficients of the individual trajectories across the 3 clusters showed that these clusters were distinct from each other: β0 of cluster 2 was significantly higher than cluster 1, which was significantly higher than cluster 3; β1 of cluster 1 was significantly higher than cluster 3, which was significantly higher than cluster 2; and β2s were not significantly different among the 3 clusters (S5 Appendix)” (see page 12 lines 269-273). In addition, as explained in the methods section, we used a combination of criteria to determine the optimal number of clusters: Pseudo F statistic (a relatively large value), R-squared value (a peak that flattens with additional clusters), Cubic Clustering Criterion (≥2), and cluster size (≥10 participants). Based on these criteria, the optimal number of clusters was 3 for both percent weight change and PHQ-9 scores. This was a data-driven approach, and hence the results may be a characteristic of the data and not generalizable. We have also added a sentence in the limitation section: “Second, the weight change and PHQ-9 score clusters may be specific to the study data. Therefore, future studies are needed to replicate the results” (see page 20 lines 408-409).

2. Further, the conclusion line in the abstract ‘Participants with failed treatment outcomes demonstrated poor engagement or progress early in the intervention’ does not, to me, seem to be clearly demonstrated. Most of those with poor engagement had no data on either weight loss or depression. And particularly for depression scores, many of those in cluster 1 never-the-less had improvements in the early period. The authors need to argue this point more exactly.

Authors’ response: We defined treatment failure using weight and SCL-20 measures objectively obtained by trained research staff at baseline, 6, and 12 months (please see our response to editor’s comment #2 as well as the measures section, pages 6-7 and lines 152-161). Participants with failed treatment outcomes fell into two categories, those with poor engagement (n=63) and those with poor progress (n=80). Although those with poor engagement had insufficient data–not no data—on either self-monitored weight or PHQ-9 scores to be included in cluster analysis, these participants could be identified early in the intervention because the majority (79%) of them discontinued the intervention by session 6 (week 8). Participants with poor progress (mainly cluster 1 participants for percent weight change or PHQ-9) could also be identified early in the intervention because they showed limited progress by 6-8 weeks compared with those in clusters 2 and 3. To further strengthen our point, we have added the dropout rate by session 6 in the poor engagement category in the abstract (see page 2 lines 48-49) and manuscript text (see page 15 lines 312-313). We have also clarified our conclusion: “Participants demonstrating poor engagement or poor progress could be identified early during the intervention and were more likely to fail treatment at the end of the intervention. This insight could inform individualized and timely optimization to enhance treatment efficacy.” 

Specific points:

3. The sensitivity analysis based jointly on weight and PHQ scores is particularly interesting since it yields a cluster whose members responded in PHQ but not in weight. This result is not evident (or at least, not emphasized) in the main cluster analysis. The concordance of this clustering result with the main analysis and the display in Table 3 might be informative.

Authors’ response: we have emphasized the concordance of the clustering results from the sensitivity analysis and the results from main cluster analyses and especially pointed out the group who responded in PHQ-9 but not in weight: “In addition, the number of participants in clusters resulting from the joint cluster analysis also showed concordance with the number of participants in Table 2. For example, the number of 33 participants with treatment progress in PHQ-9 only resulting from the joint cluster analysis (i.e., group B in S8 Appendix) was concordant with the number of 31 participants who had depression treatment progress (i.e., PHQ-9 cluster 2 and 3) but minimal weight loss (i.e., weight change cluster 1) in Table 2” (see pages 17-18 lines 350-355).

4. I suggest using, as a sensitivity analysis, clustering based not on the quarterly means but on an alternative averaging over time (e.g. 4-monthly). Further, an alternative clustering method without pre-specification of the number of clusters might help to test the robustness of the results.

Authors’ response: We did not pre-specify the number of clusters. As clarified in our response to this reviewer’s comment 1 above, we used a data-driven approach with multiple criteria. We have clarified this in the statistical analysis section: “First, the k-means method in the SAS FASTCLUS procedure without pre-specification of the number of clusters was used to group participants who had at least 1 measurement in each of the 4 quarters into clusters of individuals with similar patterns of change over time based on their 4 quarterly means. This step produced different numbers of clusters (range 2–6). Second, the optimal number of clusters was determined using a combination of criteria, including Pseudo F statistic (a relatively large value), R-squared value (a peak that flattens with additional clusters), Cubic Clustering Criterion (≥2), and cluster size (≥10 participants).[31] The optimal number of clusters was 3 for both percent weight change and PHQ-9 scores. Third, participants with percent weight change data in any 3 of the 4 quarters were assigned to their closest cluster defined by the smallest of the Euclidean distances between a participant’s 3 available quarterly means and each cluster’s means in the corresponding quarters” (see pages 7-8 lines 174-185).

We interpreted the reviewer’s comment on sensitivity analysis to suggest that we conduct a sensitivity cluster analysis using quarterly means calculated from monthly means (i.e., calculating monthly means first and then averaging the monthly means for quarterly means versus our approach which calculated quarterly means based on all available data in each quarter). Accordingly, we conducted the sensitivity analysis as suggested, and we applied the same criteria as described above for determining the optimal number of clusters. The results on percent weight change clusters were exactly the same as our original results. The results on PHQ-9 clusters still showed 3 clusters although the cluster memberships changed somewhat. Specifically 42 of 81 participants in Cluster 3 (Milder depression with treatment progress) in our method were classified in Cluster 2 (Moderate depression with treatment progress) in the sensitivity analysis—namely, the intercepts of clusters 2 and 3 were mostly affected in the sensitivity analysis. We believe this is an artifact of the “smoothing effect” of taking quarter means of monthly means in the sensitivity analysis. In accordance with the frequency of the 15 intervention sessions, PHQ-9 was administered weekly for 4 sessions, biweekly for 2 sessions, and monthly for the remaining 9 sessions. This taping-off schedule is typical of behavioral interventions. Given the session schedule, the sensitivity analysis does not differ from our analysis in calculating quarterly means for the latter 3 quarters but does affect the first quarter (and thus the intercept in particular) as it smoothed out the weekly and biweekly PHQ-9 scores in the first 2 months of the intervention and diminished the intercept and early treatment response of certain participants. As shown in the figures below, the sensitivity analysis (colored figure) moved the Cluster 2 regression line, particularly the intercept, downward. In contrast, our original method took advantage of all available PHQ-9 scores in the first quarter, more closely reflecting the real starting point and early response. Therefore, we decided to keep our method.

5. Figs. 1 and 2 display the individual trajectories but they are so small that it is difficult to get the picture. Could larger plots be shown?

Authors’ response: We have provided original-sized plots of the individual trajectories as Appendices (S2 and S4 Appendix).

6. The regression coefficient values given in appendix tables S2 and S3 differ from those in the text (pp. 9 and 10).

Authors’ response: The regression coefficient values in the text were model-based regression coefficients of each cluster (e.g., percent weight change cluster 1, 2, and 3). To verify that the resulting clusters could separate the individual trajectories within the clusters, we also provided mean (SD) of the regression coefficients of individual trajectories within each cluster in Appendices (current Appendix tables S3 and S5). ANOVA was then used to compare these regression coefficients of the individual trajectories among the 3 clusters for percent weight change and PHQ-9 separately. Therefore, the regression coefficients given in Appendix tables S3 and S5 were slightly different from those in the text and provided verification of the separation between clusters.

7. Many the ‘observed’ correlations and ‘significant’ differences are tautologous, stemming from the same or related data. For instance, the clusters were defined based on the weight and depression score changes, so it is almost inevitable that the clusters differ significantly with regard to the regression coefficients for these changes. It is similarly to be expected that the categories of treatment engagement and progress differ significantly with respect to adherence behaviors, since these behaviors were used to define the ‘no weight’ and ‘no PHQ-9’ clusters and thus the first category.

Authors’ response: We used a similar method that others have used (Babbin et al., Multivariate Behav Res. 2015, 50(1):91-108) to do cluster analysis and to verify that the derived clusters differed with regard to the regression coefficients of individual trajectories within the clusters. We have cited this paper in the statistical analysis section (page 7 line 166). We agree that percent weight cluster 0 (‘no weight’ cluster) included participants who had few weeks with self-monitored weight and PHQ-9 cluster 0 (‘no PHQ-9’ cluster) included participants who attended few sessions. In our response to the Editor’s comment 1, we underscored the potential clinical implications of these seemingly “intuitive” or “obvious” findings. To validate the distinctions of the clusters based on self-monitored weights and PHQ-9 scores over the course of the intervention, we compared weight and SCL-20 outcome data objectively obtained by trained research staff blinded to participants’ random assignment; results are shown in Table 3. 

8. Did the authors test whether the polynomial models of trajectories fitted the data well, either to the individual trajectories or to the cluster means?

Authors’ response: We have clarified in the methods section that “We also tested whether the polynomial model for each cluster fit the data well using the significance of polynomial terms, adjusted R2, and the Bayesian information criterion (BIC). For both percent weight change and PHQ-9 score change, the polynomial regression models with a quadratic term fit the data better than the ones without given the significance of the quadratic terms, higher adjusted R2, and lower BIC. Hence, the final models included both linear and quadratic terms.” (see page 8 lines 194-199). 

9. It would be interesting to investigate the possible chronological and causal relationships between weight change and depression score change: were they simultaneous or did one tend to follow the other? Or maybe no regularity can be discerned?

Authors’ response: We agree it would be interesting to test the chronological and causal relationships between weight change and depression score change. However, this is beyond the scope of this study because our data do not support the requisite design and analyses for examining temporally sensitive, causal relations. We have acknowledged that this is needed in future studies.

---

## [Decision Letter · Decision Letter 1]

27 Dec 2019

PONE-D-19-15164R1

Variability in engagement and progress in efficacious integrated collaborative care for primary care patients with obesity and depression: within-treatment analysis in the RAINBOW trial

PLOS ONE

Dear Dr. Ma,

Thank you for submitting your manuscript to PLOS ONE. After careful consideration, we feel that it has merit but does not fully meet PLOS ONE’s publication criteria as it currently stands. Therefore, we invite you to submit a revised version of the manuscript that addresses the points raised during the review process.

We would appreciate receiving your revised manuscript by Feb 10 2020 11:59PM. To enhance the reproducibility of your results, we recommend that if applicable you deposit your laboratory protocols in protocols.io, where a protocol can be assigned its own identifier (DOI) such that it can be cited independently in the future. For instructions see: http://journals.plos.org/plosone/s/submission-guidelines#loc-laboratory-protocols

We look forward to receiving your revised manuscript.

Kind regards,

Elena Barengolts, MD

Academic Editor

PLOS ONE

Reviewers' comments:

Reviewer's Responses to Questions

**Comments to the Author**

1. If the authors have adequately addressed your comments raised in a previous round of review and you feel that this manuscript is now acceptable for publication, you may indicate that here to bypass the “Comments to the Author” section, enter your conflict of interest statement in the “Confidential to Editor” section, and submit your "Accept" recommendation.

Reviewer #2: (No Response)

2. Is the manuscript technically sound, and do the data support the conclusions?

Reviewer #2: Yes

3. Has the statistical analysis been performed appropriately and rigorously? 

Reviewer #2: Yes

4. Have the authors made all data underlying the findings in their manuscript fully available?

Reviewer #2: Yes

5. Is the manuscript presented in an intelligible fashion and written in standard English?

Reviewer #2: Yes

6. Review Comments to the Author

Reviewer #2: The authors have responded in detail to the comments and made appropriate additions to the manuscript.

My remaining reservation concerns the use of significance tests on differences in trajectory parameters between clusters to prove the distinctness of the clusters (points 1 and 7). The authors replied that their method of verification was the same as that used by Babbin et al (Ref. 31):

“We used a similar method that others have used (Babbin et al., Multivariate Behav Res. 2015, 50(1):91-108) to do cluster analysis and to verify that the derived clusters differed with regard to the regression coefficients of individual trajectories within the clusters.”

As I understand Babbin’s paper, no significance tests were performed on trajectory coefficients between clusters. Cluster-mean time-series parameters were displayed (Tab. 2), but without tests. The distinctness of the clusters was verified by displaying and testing cluster mean values of baseline variables which were not involved in the clustering (Tab. 3). This is an important difference to the method of the present study, in which the tested parameters concern the trajectories of the same variable used to create the clusters.

Therefore, I maintain that these significance tests do not prove the distinctness of the clusters. Even if one created purely random data from a single distribution, clusters could be found, and such clusters would show significantly different mean values of variables derived from those variables used to create the clusters. A proper assessment of distinctness should use a method described in the literature under the key-term ‘cluster validity’. In the absence of such an analysis, claims of distinctness and significance should be toned down.

7. PLOS authors have the option to publish the peer review history of their article (what does this mean?). If published, this will include your full peer review and any attached files.

Reviewer #2: Yes: Jeremy Franklin

---

## [Author Response · Author response to Decision Letter 1]

28 Jan 2020

Authors’ response: As suggested by the reviewer, we toned down claims of the distinctness of clusters in the methods and results (lines 177-179, 233-237, and 257-261). Additionally, we emphasized the need for future validation studies with independent samples in the discussion. As described in the methods, cluster analysis was performed using the k-means method and the optimal number of clusters was decided using a combination of criteria, including Pseudo F statistic (a relatively large value), R-squared value (a peak that flattens with additional clusters), Cubic Clustering Criterion (≥2), and cluster size (≥10 participants). Although we currently do not have an independent sample to validate the 3 clusters of weight loss and the 3 clusters of depression symptoms, we stated in the discussion section that “The 3 weight loss patterns identified in the current study were similar to those found in previous weight loss studies.[34, 35]” Validation of the clusters of weight loss and depression symptoms is needed using independent samples in future studies. We emphasized this point in the discussion of the limitations (lines 387-390): “First, because of the post hoc nature of the analyses the findings need to be replicated in future studies. Second, the weight change and PHQ-9 score clusters may be specific to the study data. Therefore, future studies of independent samples are needed to verify the external validity of the results.”

---

## [Editor Report · Decision Letter 2]

31 Mar 2020

Variability in engagement and progress in efficacious integrated collaborative care for primary care patients with obesity and depression: within-treatment analysis in the RAINBOW trial

PONE-D-19-15164R2

Dear Dr. Ma,

We are pleased to inform you that your manuscript has been judged scientifically suitable for publication and will be formally accepted for publication once it complies with all outstanding technical requirements.

With kind regards,

Elena Barengolts, MD

Academic Editor

PLOS ONE
---

## [Editor Report · Acceptance letter]

8 Apr 2020

PONE-D-19-15164R2 

Variability in engagement and progress in efficacious integrated collaborative care for primary care patients with obesity and depression: within-treatment analysis in the RAINBOW trial 

Dear Dr. Ma:

I am pleased to inform you that your manuscript has been deemed suitable for publication in PLOS ONE. Congratulations! Your manuscript is now with our production department. 

With kind regards,

on behalf of

Dr. Elena Barengolts 

Academic Editor

PLOS ONE